# Extracorporeal Membrane Oxygenation-Induced Hemolysis: An In Vitro Study to Appraise Causative Factors

**DOI:** 10.3390/membranes11050313

**Published:** 2021-04-25

**Authors:** Chris Hoi Houng Chan, Katrina K. Ki, Meili Zhang, Cooper Asnicar, Hwa Jin Cho, Carmen Ainola, Mahe Bouquet, Silver Heinsar, Jo Philipp Pauls, Gianluigi Li Bassi, Jacky Suen, John F. Fraser

**Affiliations:** 1School of Engineering and Built Environment, Griffith University, Gold Coast 4222, Australia; j.pauls@griffith.edu.au; 2Critical Care Research Group, The Prince Charles Hospital, Brisbane 4032, Australia; k.ki@uq.edu.au (K.K.K.); amelia.zhang@uq.edu.au (M.Z.); casnicar199@outlook.com (C.A.); chhj98@gmail.com (H.J.C.); camen.ainola@gmail.com (C.A.); m.bouquet@uq.edu.au (M.B.); silverheinsar@gmail.com (S.H.); g.libassi@uq.edu.au (G.L.B.); j.suen1@uq.edu.au (J.S.); fraserjohn001@gmail.com (J.F.F.); 3Faculty of Medicine, University of Queensland, Brisbane 4006, Australia; 4School of Mechanical and Mining Engineering, University of Queensland, Brisbane 4067, Australia; 5Department of Pediatrics, Chonnam National University Children’s Hospital and Medical School, Gwangju 61469 42, Korea; 6School of Medicine, Griffith University, Gold Coast 4222, Australia

**Keywords:** mechanical circulatory device, blood trauma, anticoagulants, blood donor gender, flow rate, in vitro model

## Abstract

In vitro hemolysis testing is commonly used to determine hemocompatibility of ExtraCorporeal Membrane Oxygenation (ECMO). However, poor reproducibility remains a challenging problem, due to several unidentified influencing factors. The present study investigated potential factors, such as flow rates, the use of anticoagulants, and gender of blood donors, which could play a role in hemolysis. Fresh human whole blood was anticoagulated with either citrate (n = 6) or heparin (n = 12; 6 female and 6 male blood donors). Blood was then circulated for 360 min at 4 L/min or 1.5 L/min. Regardless of flow rate conditions, hemolysis remained unchanged over time in citrated blood, but significantly increased after 240 min circulation in heparinized blood (*p* ≤ 0.01). The ratio of the normalized index of hemolysis (*NIH*) of heparinized blood to citrated blood was 11.7-fold higher at 4 L/min and 16.5–fold higher at 1.5 L/min. The difference in hemolysis between 1.5 L/min and 4 L/min concurred with findings of previous literature. In addition, the ratio of *NIH* of male heparinized blood to female was 1.7-fold higher at 4 L/min and 2.2-fold higher at 1.5 L/min. Our preliminary results suggested that the choice of anticoagulant and blood donor gender could be critical factors in hemolysis studies, and should be taken into account to improve testing reliability during ECMO.

## 1. Introduction

ExtraCorporeal membrane oxygenation (ECMO) is a life-saving technology used to provide temporary cardiopulmonary venoarterial (VA) or pulmonary venovenous (VV) support for critically ill patients with refractory cardiogenic or pulmonary failure, respectively [1]. Despite these benefits, ECMO-induced hemolysis remains a challenging complication and is associated with increased morbidity and mortality [2,3]. Plasma–free hemoglobin (*pf*Hb), which seeps into the blood plasma from ruptured red blood cells (RBCs), has been commonly used to appraise the level of hemolysis. This assay has been specifically used to test the hemocompatibility of novel ECMO devices [4,5,6]. However, different animal blood or human blood used for in vitro testing [7,8], blood management (varied transport conditions) [9], and methods for hemodilution [10,11] may potentially lead to inconsistent results and further investigation is needed to ensure reliability of hemolysis measurements.

The American Society for Testing and Materials (ASTM) F1841-97 and F1830-97 standards have recommended the use of either heparin or citrate dextrose solution (ACD-A) as anticoagulants for in vitro testing of hemolysis [12,13]. Studies have emphasised alteration of cellular responses and functions using different anticoagulants [14,15]. However, the influence of anticoagulants on RBC under mechanical shear caused by extra-corporeal circulation remains unclear [12,13]. In addition, to the best of our knowledge, the effect of blood donor gender on hemolysis has never been considered and tested in comprehensive in vitro studies. Furthermore, recent studies have implied that low-flow ECMO could cause greater hemolysis, compared to high-flow conditions [16,17,18], and in various clinical situations low-flow ECMO might be required, such as during weaning from ECMO, in neonatal or pediatric applications, and during extracorporeal CO_2_ removal (ECCO_2_R), which typically operates at 0.5–2 L/min. Therefore, we designed a comprehensive in vitro study, using fresh human whole blood, to investigate the effects of different anticoagulants, blood donor gender, and high vs low blood flow conditions on hemolysis.

## 2. Materials and Methods

### 2.1. Ethics and Participant Recruitment

All experimental procedures were conducted with the approval of the Metro North Ethics Committee (HREC/17/QPCH/420) and The University of Queensland Human Ethics Research Committee (2018000105). Written informed consent was obtained from all subjects who donated blood. Eighteen healthy donors (age 30 ± 9 years) were recruited, nine of whom were female. Smokers and pregnant females were excluded from the study.

### 2.2. Preparation of ECMO Circuit Blood

Fresh human whole blood was collected (420 ± 30 mL, n = 18) by standard venipuncture into an anticoagulated blood bag containing citrate dextrose solution (66.5 mL; Macopharma, Mouvaux, France). Sodium bicarbonate (0.019 mmol/mL; Phebra, Sydney, NSW, Australia) was added to achieve a pH of 7.3 ± 0.10 and antibiotic solution (100 mg/L gentamycin; (Sigma–Aldrich, St. Louis, MO, USA) to prevent bacterial contamination [19]. For ECMO testing using citrated blood, no further adjustments were made. For heparinized blood, additional heparin (15.5 IU/mL; Pfizer, Sydney, NSW, Australia) and calcium chloride (3.57 mg/mL; Phebra, Sydney, NSW, Australia) to reverse the effect of citrate were added into the blood bag. Finally, hematocrit was adjusted to 30 ± 2% using phosphate-buffered saline (Sigma–Aldrich, St. Louis, MO, USA) to reduce variability among samples. The total time elapsed between blood collection and commencement of experiments was less than 2 h.

### 2.3. ECMO Circuit

The study was conducted in compliance with the ASTM standards for blood selection (ASTM F1830–97) [12] and in vitro blood pump evaluation (ASTM F1841–97) [13]. We used four ECMO circuits (Permanent Life Support System (PLS), Maquet CP, Rastatt, Germany) throughout our experiments (Figure 1). After each use, the ECMO circuit was thoroughly washed with saline and run in 0.6% Medizyme (Whiteley Medical Pty Ltd., Hornsby, NSW, Australia) solution rotating at a constant speed (2000 revolutions per minute (RPM)) for at least 12 h to dissolve all possible blood residue according to previously validated washing protocol [20,21]. After washing with Medizyme, 80% ethanol solution was circulated through the circuit for 1 h at 2000 RPM. The ECMO circuits were then rinsed with deionized water at a constant speed (2000 RPM) for 10 min each wash cycle (for 5 cycles).

After a thorough washout, saline was circulated for 20 min in the ECMO circuit to prime all the surfaces, and then drained prior to addition of whole blood. The four circuits were randomly assigned to different testing conditions to eliminate the risk of the individual circuit contributing to these results. Either citrated or heparinized blood was subsequently introduced into the circuit (total blood volume in the circuit, 450 ± 50 mL) and circulated for 6 h. Remaining blood was kept in a blood bag and warmed in a water bath at 37 ± 0.5 °C as a static control.

### 2.4. Hemodynamics and Data Acquisition

Prior to commencement of the experiments, the ECMO circuit was purged of any air bubbles to avoid air–blood interface. Pump speed was adjusted to 2000 RPM with temperature maintained at 37 ± 0.5 °C and enhanced gas mixture (5% CO_2_, 21% O_2_, 74% N_2_) was added at a sweep gas flow rate of 1.5 L/min. A resistance clamp was used to keep the blood flow rate constant at 4 L/min (high flow)—within the range of normal adult cardiac output. Inlet pump pressure, inlet and outlet membrane oxygenator pressures were monitored with piezoresistive pressure transducers (PX181B–015C5V; Omega Engineering, Stamford, CT, USA) and flow rate was monitored by an ultrasonic flow meter (ME10PXL1153; Transonic Systems Inc., Ithaca, NY, USA). For low flow experiments, the pump speed was gradually reduced to 980 ± 30 RPM until a blood flow rate of 1.5 L/min (low flow—weaning) was reached prior to establishing baseline. Overall hemodynamic parameters for all testing conditions were monitored and are shown in Table 1.

### 2.5. Blood Gas Analysis

Blood gas analysis was performed for each circuit condition after 5 min of circulation. pH, pCO_2_, pO_2_, bicarbonate, oxygen saturation, glucose, and electrolytes were measured using a point-of-care system (Abbott i-STAT, Chicago, IL, USA). Full blood gas examinations are shown in Table 2.

### 2.6. Hematological Parameters

Blood samples were collected at 0, 5, 60, 120, 180, 240, 300, and 360 min for a full blood count using the AcT diff™ hematology analyzer (Beckman Coulter Australia Pty Ltd., Sydney, NSW, Australia). The number of RBCs, platelets, and white blood cells (WBCs) as well as the hematocrit value (Hct) were measured.

### 2.7. Hemolysis Assay

Blood samples were collected at 5, 60, 120, 180, 240, 300, and 360 min via a sampling port in the blood reservoir. The concentration of *pf*Hb was quantified using the Harboe spectrophotometric method [22]. Briefly, plasma was isolated from blood samples by centrifugation at 15 min, 4 °C, 3000× *g*. The supernatant was then diluted 1:10 in a 0.1% sodium carbonate solution [23]. Resultant solutions were then mixed and loaded into spectrophotometric cuvettes in duplicates. Light absorbance (Abs) was measured for each cuvette at three discrete wavelengths (λ = 380; 415; 450 nm) using a UV/visible spectrophotometer (SmartSpec plus; Bio-Rad Laboratories, Hercules, CA, USA). The following equation was used determine the *pf*Hb:(1)$$pf\mathrm{Hb}\;\left(\frac{\mathrm{mg}}{\mathrm{dL}}\right)\;=(167.2\;\times \;A415\;-\;83.6\;\times \;A380\;-\;83.6\;\times \;A450)\;\times \;(\frac{1}{10})\;\times \;(\frac{1}{dilution\;in\;0.1\%\;Na_{2}CO_{3}})$$

Using the determined *pf*Hb (Equation (1)), the normalized index of hemolysis (*NIH*) was also calculated using Equation II, where Hct is the hematocrit, Q is the measured flow rate in the ECMO circuit, and t is the sampling time point:(2)$$NIH\;\left(g/100\;L\right)=\Delta \left[pf\mathrm{Hb}\right]\times loop\;volume\times \frac{100-Hct}{100}\times \frac{100}{Q\times t}$$

*NIH* is defined as grams of *pf*Hb released per 100 L of blood pumped and taking into account of hematocrit, flow rate, sampling time point, and blood volume in the loop used for standard practice for comparison of hemolytic device performance testing [24].

### 2.8. Statistical Analysis

Two-way repeated-measures analysis of variance (RM-ANOVA) with a Tukey’s post hoc multiple comparison test was performed. It was used to assess the changes in hemolysis and hematological parameters between the different anticoagulant, blood donor gender, and flow rate conditions over time. Significant differences in *NIH* were determined using a one-way analysis of variance (ANOVA). All statistical analysis was performed using Statistica, version 13.5 (TIBCO Software Inc, Palo Alto, CA, USA). *p* values ≤ 0.05 were considered statistically significant.

## 3. Results

### 3.1. Hematological Parameters

Overall hematological parameters of the samples used for all conditions at baseline are shown in Table 3. No significant differences in RBC counts, hematocrit, red blood cell distribution width (RDW), mean corpuscular volume (MCV), mean corpuscular hemoglobin (MCH), and mean corpuscular hemoglobin concentration (MCHC) were found among the different flow conditions, blood anticoagulants, and gender (Figure 2A,B). All remained constant over time and comparable to the static control. All groups but static control decreased platelet counts between 0 and 5 min of ECMO circulation (Figure 2C). There were no differences in platelet counts between high and low flow for heparinized blood. However, for citrated blood circulating at high flow, the loss of platelets was significantly greater than low flow (*p* ≤ 0.05). Between anticoagulants, heparinized blood also presented a more dramatic decrease in platelet counts, compared to citrated blood following 15 to 120 min of low flow circulation (*p* ≤ 0.05). For WBCs, no significant differences were shown when all flow and anticoagulant groups were compared against each other (Figure 2D). All groups were reported with a rapid and continuous decrease in WBC numbers over time from 5 min of ECMO circulation, except for static control.

### 3.2. Hemolysis

Regardless of ECMO flow conditions, *pf*Hb was not observed in citrated blood over time (Figure 3A). However, *pf*Hb gradually increased in heparinized blood, when compared to static control, and became significant at 240 min (low ECMO flow *p* ≤ 0.002 and high flow *p* ≤ 0.01). Between the two anticoagulants, *pf*Hb levels rose rapidly in heparinized blood, following 240 min at low ECMO flow (*p* ≤ 0.05) and 300 min at high flow circulation (*p* ≤ 0.05), contrasting the largely unchanged citrated blood. *NIH* for citrated blood at high flow and low flow were 0.0021 ± 0.00123 g/100 L and 0.0048 ± 0.00332 g/100 L, respectively (Figure 3B). *NIH* for heparinized blood at high flow (0.0246 ± 0.00732 g/100 L) was significantly lower (*p* = 0.002) than low flow (0.0794 ± 0.03387 g/100 L). In addition, *NIH* value was significantly higher in heparinized blood compared to citrated blood during low flow condition (*p* = 0.001).

As heparinized blood showed significant increases in hemolysis over time, the influence of blood donor gender was further investigated under the two flow conditions. Heparinized blood obtained from female donors had significantly less hemolysis compared to male donors following ECMO circulation (Figure 4A). The *pf*Hb levels in the circuits with male blood increased significantly at 240 min for high flow (*p* ≤ 0.05), and at 180 min for low flow (*p* ≤ 0.01), compared to female blood. Similarly, a significant higher *NIH* value was reported for blood from male donors than female donors but only for low flow (*p* = 0.0001). We also observed significant differences in *NIH* values between high flow and low flow using female (high: 0.0180 ± 0.00104 g/100 L; low: 0.0499 ± 0.00090 g/100 L) and male (high: 0.0313 ± 0.00085 g/100 L; low: 0.1089 ± 0.01597 g/100 L) blood donors (Figure 4B).

## 4. Discussions

In vitro hemolysis testing can provide valuable information about the hemocompatibility of blood-contacting medical devices and thus help to improve clinical safety. The ASTM standards for blood selection (ASTM F1830–97) and in vitro blood pump evaluation (ASTM F1841–97) recommend the use of either heparinized or citrated blood for device testing [12,13]. In addition, no recommendations regarding blood donor gender selection are currently provided. Our in vitro ECMO study shows that blood anticoagulated with heparin is associated with significantly higher rates of hemolysis than anticoagulation with citrate. In addition, we further identified that blood from male donors is more susceptible to red blood cell rupture than from females. Therefore, the choice of anticoagulants and blood donor gender are critical factors that should be taken into consideration for future in vitro hemolysis testing in ECMO and other mechanical circulatory devices.

To reflect clinical practices, we chose a flow rate of 4 L/min (within the normal adult cardiac output range, typical for full hemodynamic support using VA–ECMO) for high flow and 1.5 L/min (commonly used during weaning from VA–ECMO) for low flow measurements [25,26]. Similar to our previous work, we observed no changes in RBCs [27] but a trend of WBC loss over time [28]. However, for the platelet count, we found that citrated blood is sensitive to the flow rates over time applied in the blood loops, but heparinized blood is not. The loss of platelets during the in vitro testing of ECMO was likely due to an increase in platelet activation. Similar observation was also reported by Lu et al. [29]. In agreement with the current literature, we have also observed an increase in hemolysis when blood is exposed to low flow as compared to high flow using the present ECMO model. Specifically, we found a 3.2-fold higher *NIH* relative increase in hemolysis at low flow (1.5 L/min) compared to high flow (4.0 L/min) in heparinized blood. Similarly, Schöps et al. reported a 6.6-fold higher *NIH* relative increased hemolysis at low flow (1L/min) compared to high flow (4L/min) using a centrifugal pump DP3 [18]. Based on computational fluid dynamics (CFD), Gross-Hardt et al. demonstrated that low ECMO flow causes hemolysis due to higher recirculation within the pump, extending the blood–device contact period [17]. However, future experiments are required for validation. Nonetheless, an improved and safer design for low flow operating pumps is urgently needed to address potential cellular injury of the blood.

Systemic infusion of unfractionated heparin is the standard anticoagulation technique during ECMO. Citrate anticoagulation is rarely used clinically, with some studies reporting that its use as regional citrate anticoagulation in the ECMO circuit is feasible, safe, and effective [30,31]. Paul et al. reported that heparinized blood produced more hemolysis than citrated blood when subjected to same mechanical shear stress condition using a custom-built blood-shearing rig [32]. We conducted the first ECMO study, which reported a similar effect of anticoagulant on hemolysis. As evidenced by Makhro et al., citrate can better preserve RBC density, membrane stability, deformability, resulting in less hemolysis than heparin over 3 days of storage [9]. Therefore, blood treated with different anticoagulants show differential susceptibility to shear stress and hemolysis, and should be taken into consideration when conducting and/or comparing between hemolytic device performance studies.

Our hemolysis results showed that the female RBCs can tolerate more mechanical shear in the ECMO circuit than male RBCs before hemolysis occurs. Male RBCs consistently exhibit an increase in susceptibility to stress-induced hemolysis compared to female RBCs in response to routine cold storage [33]. Due to monthly blood loss, female blood has almost twice as many younger RBCs as male blood [34]. Younger RBCs tolerate more mechanical deformability as compared to older RBCs, due to an increased membrane stiffness in aging RBCs [35]. Previously, we demonstrated that supraphysiological shear exposure disproportionately affects older and more rigid cells within the blood, thus resulting in a “filtering” effect, whereby only the younger and more deformable cells remain [20]. To reduce the variability in future hemolysis testing, we need to explore more sensitive hemolysis assays, such as tests of sub-hemolytic damage to RBCs to evaluate blood quality check prior to when the hemolytic device performance test begins [36,37]. In future work, we will evaluate and compare pre-menopausal (more young RBCs) and post-menopausal (fewer young RBCs) female blood to confirm this result. In summary, the advantage of the reused ECMO in vitro hemocompatibility circuit study is that it is low cost and provides easy-to-evaluate hemocompatibility of ECMO, which is suitable for early prototype development. However, the potential pitfalls are hemolysis testing only and reused materials might deteriorate.

## 5. Limitations

In our study, the amount of heparin used was double the dose recommended by ASTM (F1830–19) [13]. This was related to the use of reprocessed tubing, pump, and oxygenator. In particular, tubing was depleted of the BIOLINE coating layer after rigorous cleaning. Previous studies have also used similar doses of heparin for in vitro ECMO testing [18,38]. However, we were able to maintain the physiological range of RBC, hematocrit (Figure 2), and hemodynamic (Table 1) and blood gas parameters (Table 2) among different testing conditions, consistently with our previous studies using brand new circuits [16]. We also observed similar hemolytic changes reporting a significantly greater hemolysis following circulation at low ECMO flow, compared to high flow. The ratio of current *NIH* (0.0246 g/100 L at high flow and 0.0794 g/100 L at low flow) to previous *NIH* (0.0204 g/100 L at high flow and 0.0892 g/100 L at low flow) is 1.2–fold at high flow and 0.9–fold at low flow in the ECMO circuits using heparinized blood [16]. The *NIH* differences between reused and brand-new ECMO circuits using heparinized blood for in vitro testing is negligible. Thus we demonstrated the potential for reusing ECMO circuits using heparinized blood, but only for in vitro hemolytic performance testing. Current ATSM standards for in vitro device testing recommend the use of mock circulation loops containing 450 ± 45 mL of human blood. To perform interventional investigations, multiple blood loops and large volumes of human blood are required for a single test. Therefore, the small sample size that was investigated may decrease the overall statistical significance of the study. In the future, we will investigate the minimum loop volume required (~150 mL) to adequately perform in vitro device testing, optimizing methods of volume standardization, so that we can use the same blood donor to test various conditions or devices and reaffirm these results with a bigger sample size.

## 6. Conclusions

Our preliminary results suggested that the choice of anticoagulant and donor blood gender could be critical factors when conducting in vitro hemolysis testing. Therefore, we recommend that the American Society for Testing and Materials standards should consider including these factors to improve reproducibility during in vitro testing of ECMO equipment. In addition, we demonstrated that ECMO circuit could be reliably recycled for in vitro hemolysis testing, following our rigorous washing protocol.

## Figures and Tables

**Figure 1 membranes-11-00313-f001:**
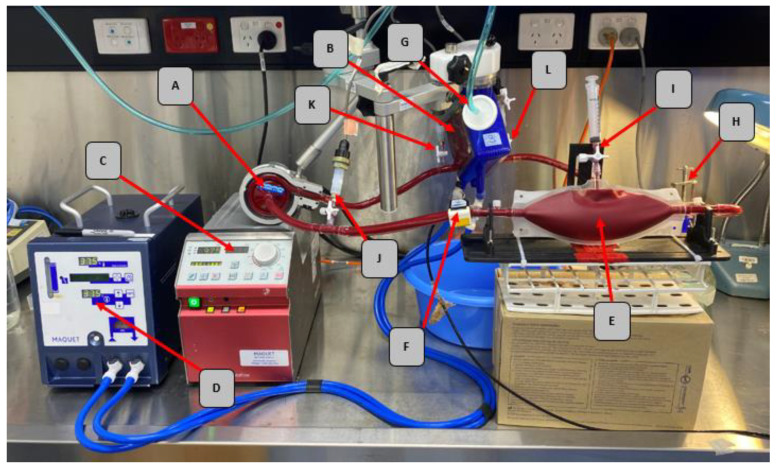
Schematic of an in vitro circuit of extracorporeal membrane oxygenation (**A**) ROTAFLOW centrifugal pump (**B**) MAQUET PLS-i oxygenator (**C**) ROTAFLOW pump console for flow regulation (**D**) HU 35 Heater Unit (**E**) Blood reservoir (**F**) Flow sensor (**G**) Sweep gas line inlet (**H**) Resistance clamp (**I**) Sampling port (**J**) Pump inlet pressure sensor (**K**) Oxygenator inlet pressure sensor and (**L**) Oxygenator outlet pressure sensor.

**Figure 2 membranes-11-00313-f002:**
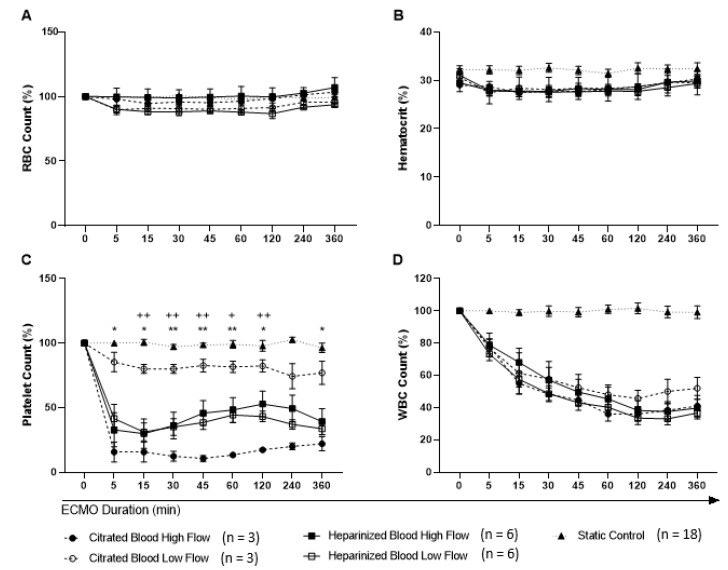
Changes in hematological parameters following ECMO circulation over time. (**A**) RBC count (**B**) hematocrit levels, and (**D**) WBC count except static control showed no significant change between the different flow conditions. In contrast (**C**) platelet count showed significant change between the two flow rates in citrated blood (* *p* ≤ 0.05 and ** *p* ≤ 0.01) as well as between heparinized and citrated blood in low flow settings between 15 and 120 min (+ *p* ≤ 0.05 and ++ *p* ≤ 0.01). Data are presented as mean ± SEM. RBC: red blood cell count; WBC: white blood cell count.

**Figure 3 membranes-11-00313-f003:**
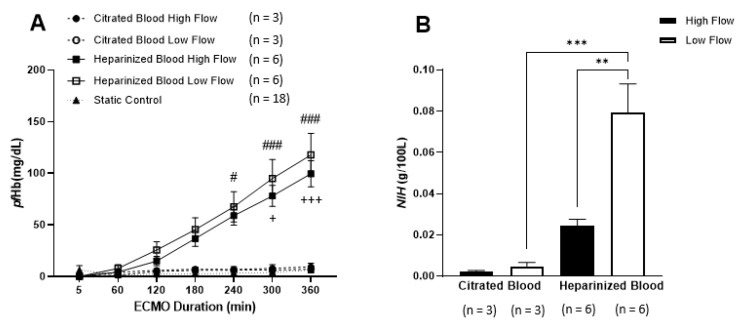
Evaluation of hemolysis between different flow rates, anticoagulants and a static control. (**A**) The levels of plasma free hemoglobin (*pf*Hb) following exposure to varying conditions over 360 min of in vitro ECMO circulation were compared. Regardless of flow conditions, *pf*Hb increased over time in heparinized blood compared to static control, whereas in citrated blood hemoglobin levels remained similar. High flow heparinized blood vs citrated blood + *p* ≤ 0.05 and +++ *p* ≤ 0.001; low flow heparinized blood vs citrated blood # *p* ≤ 0.05 and ### *p* ≤ 0.001. (**B**) Normalized indexes of hemolysis (*NIH*) were also calculated. *NIH* significantly increased in heparinized blood compared to citrated blood during low flow conditions. Additionally, *NIH* was decreased in high flow compared to low flow settings in heparinized blood. ** *p* ≤ 0.01 and *** *p* ≤ 0.001. Data are presented as mean ± SEM.

**Figure 4 membranes-11-00313-f004:**
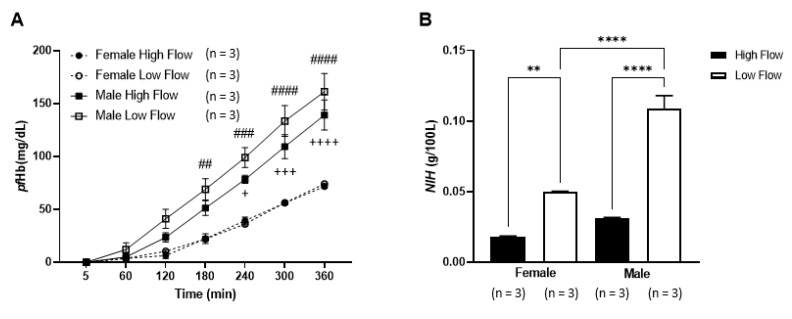
Evaluation of hemolysis between blood donor gender and flow rates in circulating heparinized blood. (**A**) The levels of plasma free hemoglobin (*pf*Hb) from female and male donor blood following exposure to high or low flow were compared. Regardless of flow conditions, female blood has significantly lower hemolysis compared to male. ## *p* ≤ 0.01, ### *p* ≤ 0.001 and #### *p* ≤ 0.0001 indicate significant differences between female blood and male at blood low flow. + *p* ≤ 0.05, +++ *p* ≤ 0.001 and ++++ *p* ≤ 0.0001 indicate significant differences between female blood and male blood at high flow. (**B**) Normalized index of hemolysis (*NIH*) were also calculated. Within gender, *NIH* is significantly increase during low flow when compared to high flow. Additionally, in a low flow settings, male blood showed a significantly higher *NIH* compared to female blood. ** *p* ≤ 0.01 and **** *p* ≤ 0.0001. Data are presented as mean ± SEM.

**Table 1 membranes-11-00313-t001:** Hemodynamic Parameters Flow Regimes of the ECMO Blood Circulation Loops.

Hemodynamics Parameters	Citrated Blood, High Flow (n = 3)	Citrated Blood, Low Flow (n = 3)	Heparinized Blood, High Flow (n = 6)	Heparinized Blood, Low Flow (n = 6)
Flow Rate (L/min)	4.0 ± 0.01	1.5 ± 0.01	4.0 ± 0.02	1.5 ± 0.01
Pump Speed (RPM)	2004 ± 3	962 ± 47	2003 ± 4	989 ± 16
Inlet Pump Pressure (mmHg)	12 ± 0.2	–1 ± 1.2	–14 ± 1.5	–3 ± 0.9
Inlet Oxygenator Pressure (mmHg)	83 ± 0.4	18 ± 1.7	83 ± 2.5	19 ± 0.5
Outlet Oxygenator Pressure (mmHg)	61 ± 5.1	11 ± 0.8	63 ± 5.1	10 ± 1.3
Pressure Drop Across the Oxygenator (mmHg)	22 ± 4.9	7 ± 0.9	20 ± 2.9	9 ± 1.2

Data is presented as mean ± standard deviation.

**Table 2 membranes-11-00313-t002:** A blood gas was evaluated for each circuit condition after 5 min of circulation.

Blood Gas Parameters	Citrated Blood, High Flow (n = 3)	Citrated Blood, Low Flow (n = 3)	Heparinized Blood, High Flow (n = 6)	Heparinized Blood, Low Flow (n = 6)
pH	7.3 ± 0.02	7.3 ± 0.01	7.3 ± 0.05	7.3 ± 0.06
pCO_2_ (mmHg)	34.9 ± 0.95	34.2 ± 0.09	34.8 ± 0.62	34.4 ± 1.17
pO_2_ (mmHg)	126.8 ± 12.41	126.6 ± 18.50	133.8 ± 5.44	131.2 ± 6.22
Bicarbonate (mmol/L)	16.8 ± 1.76	16.6 ± 0.27	18.5 ± 2.29	15.9 ± 2.53
Oxygen Saturation (%)	98.6 ± 0.51	98.5 ± 0.71	98.8 ± 0.50	98.6 ± 0.55
Glucose (mmol/L)	11.6 ± 2.22	14.9 ± 5.19	14.1 ± 2.18	11.4 ± 1.82
Potassium (mmol/L)	2.2 ± 0.10	2.5 ± 0.37	2.4 ± 0.17	2.5 ± 0.32
Sodium (mmol/L)	153.5 ± 2.21	153.0 ± 8.49	159.0 ± 4.00	159.6 ± 1.14
Calcium (mmol/L)	<0.25	<0.25	>2.50	>2.50

Data is presented as mean ± standard deviation.

**Table 3 membranes-11-00313-t003:** Baseline hematology data (t = 0 min) for each experimental mock circulation loop condition.

Parameter	Unit	Citrated Blood, High Flow	Citrated Blood, Low Flow	Heparinized Blood, High Flow	Heparinized Blood, Low Flow	Static Control
Red blood cell (RBC) count	10^6^⸱µL^−1^	3.23 ± 0.04	3.52 ± 0.19	3.44 ± 0.77	3.51 ± 0.49	3.64 ± 0.41
White blood cell (WBC) count	10^3^⸱µL^−1^	4.3 ± 0.97	4.2 ± 0.56	4.7 ± 1.80	4.2 ± 0.50	4.7 ± 1.80
Mean corpuscular volume (MCV)	fL	90.7 ± 0.81	88.8 ± 1.76	89.0 ± 4.74	89.0 ± 6.24	88.6 ± 4.88
Mean corpuscular hemoglobin (MCH)	pg	29.7 ± 0.75	28.5 ± 0.95	29.0 ± 1.36	29.4 ± 1.65	29.4 ± 1.62
Mean corpuscular hemoglobin concentration (MCHC)	g⸱dL^−1^	330.3 ± 2.52	322.0 ± 4.36	326.2 ± 7.03	330.83 ± 6.79	328.0 ± 6.21
Red blood cell distribution width (RDW)	%	12.0 ± 0.67	12.9 ± 0.74	12.7 ± 0.91	12.5 ± 0.31	12.6 ± 0.64
Platelet count	10^3^⸱µL^−1^	189.7 ± 35.22	183.7 ± 54.24	190.2 ± 39.1	180.5 ± 43.65	187.5 ± 47.12

Data is presented as mean ± standard deviation.

## Data Availability

The datasets generated and/or analyzed during the current study are available from the corresponding author upon reasonable request.

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
