# Peer review of "Extracorporeal Membrane Oxygenation-Induced Hemolysis: An In Vitro Study to Appraise Causative Factors"

_membranes, 2021, doi:10.3390/membranes11050313_

Round 1

Reviewer 1 Report

Lines 39-41: Could the authors please provide examples of the main factors that could be considered responsible for the inconsistent results and how the new method described in the manuscript could help managing those?

Discussion: Could the authors please summarise the advantages and the potential pitfalls of the described method preferably in a table

Author Response

Point 1: Lines 39-41: Could the authors please provide examples of the main factors that could be considered responsible for the inconsistent results and how the new method described in the manuscript could help managing those?

Response Point 1: All mentioned factors (heterogeneity of blood sources, blood management, methods for hemodilution) are responsible for the inconsistent results. However, poor reproducibility remains a challenging problem, due to several unidentified influencing factors. Therefore, we tested re-used ECMO circuit for in vitro testing and identified other possible influencing factors such as anticoagulants and gender blood donor too.

The American Society for Testing and Materials (ASTM) F1841-97 and F1830-97 standards serve as an important platform to standardise them, identify and recommend any “influencing factors” which could potentially affect the hemolysis testing outcome so that a fair blood trauma testing can be compared between devices such as ECMO.

In order to help managing these influencing factors, we added that “Therefore, we recommend that American Society for Testing and Materials standards should consider including these factors to improve reproducibility during in vitro testing of ECMO equipment”(lines 282-284). 

Point 2: Discussion: Could the authors please summarise the advantages and the potential pitfalls of the described method preferably in a table

Response Point 2: We thank the authors for their comment, and have added “In summary, the advantage of reused ECMO in vitro hemocompatibility circuit study provides a low cost and easy to evaluate hemocompatibility of ECMO, which is suitable for early prototype development. However, the potential pitfalls are hemolysis testing only and reused materials might deteriorate.” (Lines 262-264) to summarise the advantages and the potential pitfalls of the current method.

Reviewer 2 Report

The present study reports on anticoagulant-mediated and sex-specific differences in hemolysis with ECMO under both high and low flow conditions. The investigators find that citrated blood is protected from hemolysis compared to heparinized blood and that hemolysis is lower in female blood than male blood. Both are new findings and are of clinical interest. There are some additional pieces of data (minor) that should be included in the Results section or should be addressed as a limitation of the study. Otherwise, this the study is well done and well presented.

The present study reports on anticoagulant-mediated and sex-specific differences in hemolysis with ECMO under both high and low flow conditions. The investigators find that citrated blood is protected from hemolysis compared to heparinized blood and that hemolysis is lower in female blood than male blood. Both are new findings and are worthy of clinical consideration. The study is well done. A few minor comments.

  1. Line 39 – what is meant by heterogeneity of blood sources and different blood species?
  2. As the ECMO circuits are reused, is there any difference in the material properties (i.e, stiffness) of the tubing after reuse, cleaning, etc.?
  3. The authors correctly point out that female blood is more likely to have newer more deformable RBCs because of the menstrual cycle. Do you have any idea at what stage of the cycle the women were at when blood was collected? Any post-menopausal women included?
  4. As a corollary to 3, it would be helpful to include a table with baseline CBC and include red cell distribution width (RDW), reticulocyte count, etc to support your comments about deformability of female cells. Would also add ABO group.
  5. Can you comment on the high sodium levels, low potassium levels, high glucose levels, and low calcium levels (in the citrated blood) measured by ABG in the circuits? This could also affect RBC stiffness.

Author Response

Point 1: Line 39 – what is meant by heterogeneity of blood sources and different blood species?

Response Point 1: We amended to “different animal blood or human blood used for in vitro testing”.

Point 2: As the ECMO circuits are reused, is there any difference in the material properties (i.e, stiffness) of the tubing after reuse, cleaning, etc.?

Response Point 2: Thanks. It is a valid point. We agree that reused material (whole circuit) properties (e.g. stiffness of the material properties) might deteriorate. Therefore, we highlighted this in following sentences (Lines 262-264), “In summary, the advantage of reused ECMO in vitro hemocompatibility circuit study provides a low cost and easy to evaluate hemocompatibility of ECMO, which is suitable for early prototype development. However, the potential pitfalls are hemolysis testing only and reused materials might deteriorate.” In addition, we have now clarified this point that “the four circuits were randomly assigned to different testing conditions, to eliminate the risk of the individual circuit contributing to these results (lines 86-87)”.

Point 3: The authors correctly point out that female blood is more likely to have newer more deformable RBCs because of the menstrual cycle. Do you have any idea at what stage of the cycle the women were at when blood was collected? Any post-menopausal women included?

Response Point 3: Unfortunately, we did not collect information regarding stage of the menstrual cycle from female volunteers at time of blood collection. However, we know that the nine healthy females (age 30 ± 9 years) recruited were premenopausal. Therefore, no postmenopausal women included in this work. Reviewer has raised a very interesting point (pre-menopausal vs post-menopausal). We have added “In future work, we will evaluate and compare pre-menopausal (more young RBCs) and post-menopausal (fewer young RBCs) female blood in order to confirm this result” (lines 260-261).

Point 4: As a corollary to 3, it would be helpful to include a table with baseline CBC complete blood count and include red cell distribution width (RDW), reticulocyte count, etc to support your comments about deformability of female cells. Would also add ABO group.

Response Point 4: We have added a table with baseline CBC complete blood count (Table 3) (line 165) and include red cell distribution width (RDW), reticulocyte count, etc. A statistical analysis was performed to compare RDW, MCV, MCH and MCHC between male and female. No significant differences we observed between genders.

Unfortunately, we did not perform the ABO blood test, therefore, we cannot add this information.

Point 5: Can you comment on the high sodium levels, low potassium levels, high glucose levels, and low calcium levels (in the citrated blood) measured by ABG in the circuits? This could also affect RBC stiffness.

Response Point 5: The level of sodium, potassium, glucose are similar between citrated and heparinised blood under both high and low flow conditions. Therefore, the differences in hemolysis between citrate and heparin are unlikely due to these parameters.

However, low calcium levels in the citrated blood may as the reviewer suggested played a role in RBC stiffness, which resulted in less hemolysis. As past evidence have shown that “As evidenced by Makhro et al., work, citrate can better preserve RBC density, membrane stability, deformability, resulting in less hemolysis than heparin over 3 days of storage” (lines 246-248)

Reviewer 3 Report

This is a small experimental study with just 18 donors. There are not enough data to support a strong conclusion.

Author Response

Point 1: This is a small experimental study with just 18 donors. There are not enough data to support a strong conclusion.

Response Point 1: We added "However, the small sample size may decreased the statistical power of the study." into the limitation section - lines 276-277.

We agree reviewer's point and have tuned down the conclusion statement as "Our results suggested that the choice of anticoagulant and donor blood gender could be critical factors when conducting in vitro hemolysis testing. " - lines 280-281. 

Round 2

Reviewer 3 Report

I have no further comments regarding the manuscript. The sample size is very small and cannot support a strong conclusion. These are only preliminary data.

Author Response

Point 1: I have no further comments regarding the manuscript. The sample size is very small and cannot support a strong conclusion. These are only preliminary data.

Response Point 1: We added “Current ATSM standards for in vitro device testing recommend the use of mock circulation loops containing 450 ± 45 mL of human blood. To perform interventional investigations, multiple blood loops and large volume of human blood are required for a single test. Therefore, small sample size was investigated which may decrease the overall statistical significance of the study. In future, we will investigate the minimum loop volume required (~150 mL) to adequately perform in vitro device testing, optimizing methods of volume standardization, so that we can use same blood donor to test various conditions or devices and reaffirm these results with a bigger sample size.  “ – lines 273-278. 

We have also amended that “Our preliminary results suggested that the choice of anticoagulant and donor blood gender could be critical factors when conducting in vitro hemolysis testing.” in the conclusion section – lines 281-282.

We have also amended “Our preliminary results suggested that the choice of anticoagulant and blood donor gender are could be critical factors in hemolysis studies, and should be taken into account to improve testing reliability during ECMO.” in the abstract section – lines 26-28.
